# Contribution to the knowledge of early geotechnics during the 20th century: Laurits Bjerrum

Gonzalo Guillán-Llorente, Belén Muñoz-Medina, Antonio Lara-Galera, Rubén Galindo-Aires

Technical School of Civil Engineering, Universidad Politécnica de Madrid, Madrid, 28040, Spain,

*Correspondence to*: Belén Muñoz-Medina (mariabelen.munoz@upm.es)

**Abstract.** The founder of Soil Mechanics, Karl Terzaghi, took the initiative in 1954 to contact the Danish Engineer Laurits Bjerrum to meet him. Terzaghi wanted to meet the engineer who had written a paper on the stability of the unusual Norwegian quick clays at the European Slope Congress in
Stockholm. Bjerrum was 36 years old at the time, had a PhD and was already director of the NGI (Norges Geotekniske Institutt - Norwegian Geotechnical Institute). From his position as director of the NGI, he was actively involved in many varied consultancies, placing great value on the continuous interaction between practice and research. Bjerrum's strategy for establishing the NGI came from the experience of other research centres such as the BRS (Building Research Station) in Great Britain and
Imperial College London. In addition, having lived through the Nazi occupation of Denmark, he was predisposed against the misuse of authority and established an open structure for the Institute from its inception. Bjerrum was in close contact with the Norwegian Institution of Technology, and in 1952, he succeeded in getting Soil Mechanics incorporated as a compulsory subject in the civil engineering degree. Subsequently, in 1960, the Chair of Soil Mechanics and Foundation Engineering were
established. The first laboratory of this chair was equipped with material donated by the NGI. Bjerrum died young (54 years old) but he had built an excellent reputation through his work at the NGI and his contributions to the International Congresses, where he maintained a close relationship with the significant figures in geotechnics: Terzaghi, Skempton, Peck and Casagrande. He made regular trips to the USA, where he was a visiting professor at M.I.T. (Massachusetts Institute of Technology) and
received the highest international decorations.
**1 Origins**

Laurits Bjerrum (Farsø, Denmark, 6 August 1918; London 27 February 1973) was the eldest son of Chresten A. Bjerrum, a Danish surgeon, head of surgery at the Farsø district hospital in the northern Danish peninsula of Jutland, and Henrietete Krag Hansen, married in 1917, a year before Laurits was

born, (Bjerrum, 1954a). His brother Jørgen was born a year later, in 1919. Laurits and Jørgen's father, who was not only a doctor but also a skilled investor, died in 1928 at the age of 61, leaving his family well off. Laurits was 10 years old, when his father passed away, so he did not have much influence on the young Laurits.

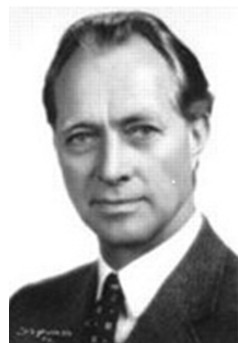


**Figure 1: Laurits Bjerrum, (Guillán Llorente, 2015).**

Shortly before his father's death, the Bjerrum-Hansen family had moved to Ribe, in the southern part of Jutland, Denmark, where the Bjerrum family originated. Laurits studied at the Ribe Katedralskole,

graduating in 1936, and was known for his frequent pranks with his friend Jannik Ipsen and his love of watercolour painting. One of his pranks, as told by his friend Jannik Ipsen, was to summon the dignitaries of the city of Ribe to the best hotel to hear the Dean talk about "MacAdam's mission", which was pure imagination. The idea came to him from the new method of road construction, macadamization. It created quite a stir when everyone found out that it was a joke. For this, he was

almost expelled from the institute.

Laurits then went to Copenhagen where he graduated in Civil Engineering at the Technical University of Denmark (DTU, its Danish acronym) in January 1941, during the second year of the German occupation.

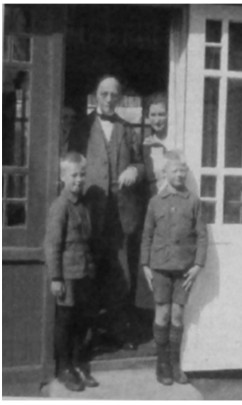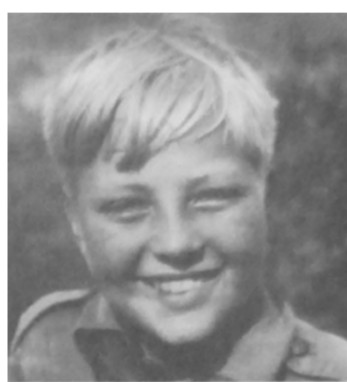

**Figure 2. Laurits Bjerrum with his parents and brother at the hospital in Farsø, 1927 and Laurits as a teenager in Ribe, 1933. (Bjerrum, Ch. A, 2003)**

While attending the Technical University of Denmark, Bjerrum worked at the Laboratory of Harbour Works and Foundations between 1941 and 1942 assisting Dr. Bretting with his research projects in fluid

mechanics and marine structures at DTU. Shortly after finishing his studies, Laurits married Gudrun, the daughter of a Danish merchant navy captain, in 1941, with whom he had three children, Chresten (1942), Annette (1947) and Susanne (1953).

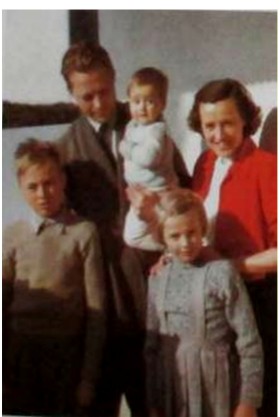

**Figure 3. The Bjerrum family in 1954. (Bjerrum, G., 2003)**

## 2 Professional start in Engineering

During the four years between 1942 and 1946 Laurits was employed by Dr. Christen Ostenfeld in the Soil Mechanics and Foundation Engineering section of the engineering group Chr. Ostenfeld & W. Jønson (currently renamed COWIconsult A/S) of which he became, at 23, the head of the geotechnical department although initially, it consisted of him alone. During the first part of this period, he also continued to work for the Technical University in the laboratory for harbour works and foundations. Chr. Ostenfeld & W. Jønson was at the time a small company with only ten enthusiastic employees led by Ostenfeld, who was clear about the importance of soil investigation in solving foundation problems. During his time at Chr. Ostenfeld & W. Jønson, Laurits Bjerrum had the opportunity to work almost exclusively on soil investigation (or "drilling in the province" as he used to say) and soil mechanics. The works, in those times of necessity due to the Second World War, were of modest scale but of diverse nature and often with complicated technical problems. At Chr. Ostenfeld & W. Jønson, he had the opportunity to work as a consultant assigned to the project for a new highway bridge over the Little Belt between Jutland and the island of Fyn. On that project, he worked in a team with Jørgen Brinch Hansen, who would later become a professor of geotechnics at DTU in 1955 and director of the Danish Geotechnical Institute (DGI) for 15 years.

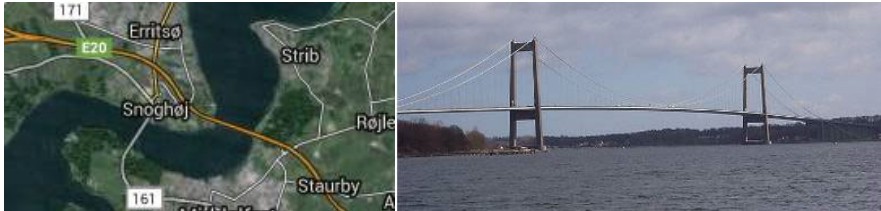

**Figure 4. The highway bridge over Little Belt (© Google Maps and Wikipedia)**

In his spare time, and with permission from the Danish Railways, Bjerrum studied the stability of
railway embankments in various parts of Denmark where the ground conditions were difficult, enabling
him to gather the first data for his subsequent doctoral thesis. Bjerrum was interested in the Jutland
main line and, in particular, in a section along the Vejle Fjord, with scarps about 50 metres high in
Tertiary micaceous clays; this was an extremely unstable area that had caused many problems for the
Danish Railways (DSB) over the years.

It was probably during that work that Bjerrum met the Danish geologist Ellen Louise Mertz, as Mertz
had worked for the Danish railways since the construction of the first bridge over the Little Belt in the
1930s. The relationship between Bjerrum (Soil Mechanics) and Mertz (Engineering Geology) had a
great influence on the development of geotechnical engineering in Denmark. Mertz, who also worked in
other Scandinavian countries, had a good reputation for excellent professional judgment. This
reputation had an important impact on Bjerrum's professional development, as we shall see below.

During his review of publications on slope slides, Bjerrum came across the name of Professor R.
Haefeli of the Research Institute for Hydraulic Engineering and Civil Engineering of the Swiss Federal
Institute of Technology (ETH its German acronym) was particularly interested in the work by Haefeli
on the mechanics of snow and ice and its close connection with soil mechanics. In particular, Bjerrum
wanted to investigate the problem of "creeping" in snow mechanics, which led the young Bjerrum to
leave Denmark for Zurich in 1947.

According to Bjerrum's colleagues at Chr. Ostenfeld & W. Jønson, Gregers Vefling and Aksel G.
Frandesen, the Swiss connection was provided through Chresten Ostenfeld, who as a young man, had
worked as an assistant at the ETH.

**3 Swiss Federal Institute of Technology (ETH).**

Laurits Bjerrum joined ETH Zurich as a collaborating scientist to Professor Eugene Meyer-Peter, Director of the Research Institute for Hydraulic Engineering and Civil Engineering at ETH, joining the team in the soil mechanics and hydraulics laboratory.

Within the ETH civil engineering laboratory, Eugene Meyer-Peter founded the soil mechanics
laboratory in 1935 and entrusted Robert Haefeli with its management. In its early days, the laboratory was mainly involved in the production of so-called "expert judgements" and research, and, despite its small staff, it was well regarded.

Unfortunately, at that time R. Haefeli was very ill, so the young Bjerrum occupied himself in the preparation of judgement reports and laboratory work. Laurits took the opportunity to familiarise
himself with rock mechanics and the practical problems of the different soils in Switzerland, which were very different from those in Denmark.

At the ETH, Bjerrum became aware of the importance of the continuous interaction between practice and research. For the rest of his professional life, Bjerrum remained faithful to this principle.

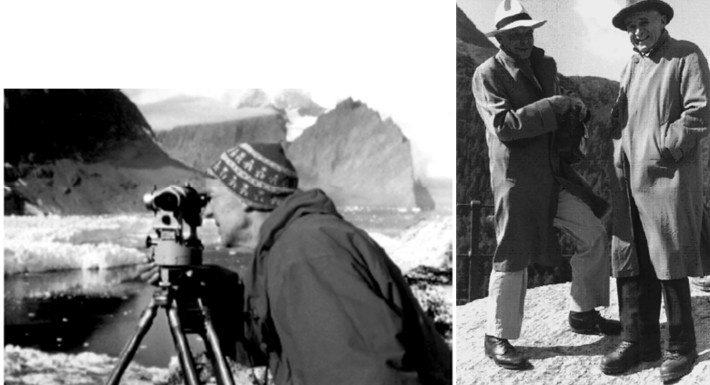

**Figure 5. (Left) Robert Haefeli (photo Huder, J.) (Right) Karl Terzaghi with Eugene Meyer-Peter (in dark hat) on the Maloja Pass in 1953 (photo Huder, J., 1979)**



Bjerrum undertook frequent professional trips, which allowed him to familiarise himself with various
areas of Switzerland. He dealt with different aspects of soil mechanics, including but not exclusively,
foundations, settlement problems, landslides, support for road construction, in the various soils of
Switzerland. Some of these trips were made in the company of Jachen Huder, with whom he developed
a close friendship and with whom he became a professor at the ETH.

One of the main concerns of the Zurich Research Institute was dams of loose materials. The most
important dam of all was the Marmorera loose material dam at Oberhalbstein in Canton Grisons,
Switzerland. It was the first Swiss loose material dam of these dimensions: height 91 m, length 400 m,
reservoir volume 60 hm$^3$, reservoir area 1.41 km$^2$. The dam was visited, years later, by the attendees of
the 3rd International Congress on Soil Mechanics and Foundation Engineering held in 1953 in Zurich.

During the construction of the Marmorera dam, Bjerrum was responsible for the installation and
operation of the first modern construction laboratory in Switzerland. Theoretical knowledge could be
directly applied here and corrections to the design were directly verified by field measurements.
Bjerrum researched the compaction of coarse granular soils, which was published in German in the
technical journal Strasse und Verkehr in 1952, (Bjerrum, 1952). Bjerrum spoke fluent German, French
and English, and some Italian, but spoke very little Norwegian despite having lived in Norway for more
than 30 years, (Bjerrum, 2003).

In 1948, Bjerrum attended the second International Congress on Soil Mechanics and Foundation
Engineering (ICSMFE) in Rotterdam, where he met, among others, the English geotechnician Alec
Skempton.

In 1950, a large triaxial vacuum apparatus was developed to test the shear strength of the material
placed on site (with a diameter of 0.6 m and a height of 1.2 m). Extensive experiments were carried out
in the Marmorera field laboratory, including tests on samples with a diameter of about 50 cm and a
height of 100 cm. The annual displacement measurements carried out over 20 years, show an almost
elastic behaviour of the Marmorera loose material dam, which is evidence of the satisfactory
compaction of the dam material.

**4 Norwegian Geotechnical Institute (NGI).**

In 1950, the Norwegian engineer Sverre Skaven-Haug and the Danish geologist Ellen Louise Mertz informed Bjerrum that the newly established Norwegian Geotechnical Institute (NGI) in Oslo was looking for a director and recommended that he apply for the position, which he did.

The NGI was and is one of several research centres administered by the Royal Norwegian Council for
Scientific and Industrial Research (NTNF). This semi-official Council (NTNF) was established in 1946 by an act of the Norwegian parliament to promote scientific and industrial research and to ensure that the results obtained were used for the benefit of the country. The Council took steps to establish new research institutions, in research fields with inadequate research means and the NGI was one of them.

One man stands out from that beginning, Olav Folkestad, who was responsible for the initial ideas that
ultimately led to the creation of the Institute. Folkestad was the chairman of the Geotechnical Committee (the germ of the NGI), with Sverre Skaven-Haug and R. Gran Olsson representing the Norwegian Institute of Technology (NTH) as representatives of the Norwegian Railways.

Olav Folkestad, a renowned civil engineer mainly engaged in consultancy, realised early on that Norway needed a geotechnical research institute; and campaigned vigorously until he succeeded. From
1947 to 1952 Folkestad chaired the committees responsible for the objectives for the establishment of such an institute.

At the request of the Norwegian Institute of Technology, the NTNF decided to establish a geotechnical laboratory, under the management of O. Kummeneje, who was replaced in 1951 by Nilmar Janbu for a short time, and later by Rolf Christian Vold. The Norwegian Institute of Technology and this new
geotechnical laboratory were commissioned to prepare the field investigation for the rapid transit system in Oslo, which included large sections in deep excavations and tunnels through soft clays and rocks, including shales susceptible to swelling. The Geotechnical Committee received its first consultancy work in a brief letter from the manager of the Oslo Underground Railway Planning Office, J. Vogt-Nilsen, specifying that, as the scope of work was not determined, the work would be carried out
on a "gentlemen's agreement" basis. Rarely do major works of this kind owe their success to such simple contracts. A comparable contract was the Chicago underground contract involving Terzaghi and Peck.



In 1950 a permanent geotechnical secretariat was created, and, in the same year, an advertisement was circulated in Scandinavia seeking a director for what was to be known as the NGI, to which, as
mentioned above, Bjerrum responded.

At that time Laurits Bjerrum was a young Danish civil engineer employed at EHT in Zurich, totally unknown to Olav Folkestad and unaware of the geotechnical role that Norway would play in the future.

The missing link between Olav Folkestad and Laurits Bjerrum was, the Norwegian engineer Sverre Skaven-Haug, a member of the Geotechnical Committee, and the aforementioned Danish geologist
Ellen Louise Mertz, who knew both people well. It was they who recommended Bjerrum for the advertised position in Norway.

Olav Folkestad visited Bjerrum in Zurich and invited him to spend a few weeks in Norway to get to know the people and the geotechnical problems in Norway. Norway was soon convinced that Bjerrum was the right person for the job.

Laurits arrived in Oslo on 2 October 1951, at the age of only 33, to take up the new position of director and head of what was to become officially known, in January 1953, as the NGI, a position he held until his early death.

Olav Folkestad himself picked up the Bjerrum couple from the ferry that brought them from Copenhagen to Oslo and took them to their new home in Lillevann, in the Holmenkollen hills. In later
years, they would move a little closer to Oslo, to Trosterudveien.

When Laurits Bjerrum arrived in Norway, the Norwegian Geotechnical Institute existed only in a few people's minds. The staff consisted of three people, engineer Rolf Christian Vold who headed the office, engineer Ove Eide responsible for the technical office and technician Arild Andresen, all working in the same office, a simple wooden structure in Blindern.

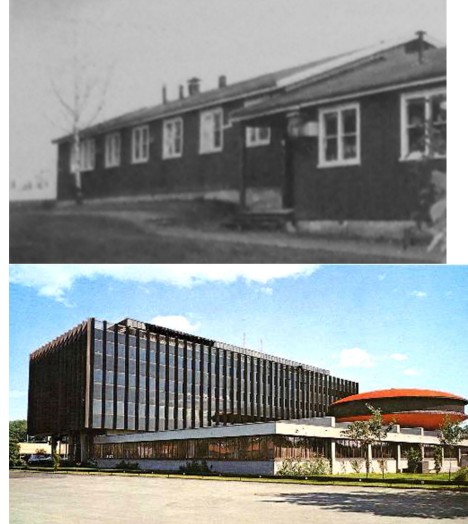


**Figure 6. (Left) Early NGI facilities at the university campus in Blinden, Oslo. (Right) Current NGI facilities, Oslo. (Von, R.C., 2003)**

Even though the principles of soil mechanics were being successfully put into practice by engineers in
Norway and circumstances at the time were very favourable to the creation of a geotechnical research centre, Bjerrum understood this was a unique opportunity. Although there was a national need for the services of such an institution, there was little or no precedent set in Norway on how to go about it, but neither was there any precedent that prevented or limited it. Equally important was the fact that, from the very beginning, Bjerrum had the enthusiastic support of the authorities who gave him financial
backing and almost complete freedom to select his staff and develop his ideas for the Institute. What started with one office and three people in 1951 grew rapidly and steadily under his leadership to become the well-known Norwegian Geotechnical Institute, NGI.

NGI is today the natural result of the efforts of many people from all walks of life and especially of Olav Folkestad and Laurits Bjerrum, who with their own hands and starting from nothing, skilfully put
together the parts and people that today form the Institute.

Bjerrum, from his position at the NGI, had an important influence on the inclusion of soil mechanics and foundation engineering as a compulsory subject in the civil engineering course at the Norwegian





Institute of Technology (NTH). The first courses in Soil Mechanics between 1952 and 1960 were a joint effort of the University and several people at NGI, including Bjerrum, until the Chair of Geotechnics was established in 1960.


The first laboratory of this new chair was equipped with equipment donated by the NGI. At the same time, an agreement was signed to facilitate doctoral theses at the NGI and to ensure up-to-date teaching using visiting professors (the NGI Engineers), which enabled Laurits to establish a reputation as an outstanding professor.

During this time Bjerrum completed his research for his doctoral thesis, "Theoretical and experimental investigations of shear stresses in soils", (Bjerrum, 1954a), with which he obtained his doctorate in technological sciences from ETH Zurich in 1954. Doctoral thesis supervised by Prof. Meyer-Peter, E. and Prof. Haefeli, R., for which he received the degree Dr. Sc. Technology from the Federal Institute of Technology (ETH) Zurich. This confirmed the beginning of his interest in this fundamental topic, an

interest that was to culminate in his outstanding and advanced report on soft clays. He was due to present this report in a paper at the 8th International Congress on Soil Mechanics and Foundation Engineering in Moscow in 1973, (Bjerrum, 1973), (Bjerrum & Berre, 1973) but his sudden death in the same year from a heart attack prevented this presentation.

To prevent the NGI from stagnating, he ensured that, in addition to research, the NGI's objectives

included consultancy, thus keeping the entire staff up to date on construction procedures. In addition, he encouraged international cooperation, attendance at conferences and the exchange of information with universities and research institutions abroad. This also included trips to other centres abroad, and he encouraged his colleagues abroad to come to Norway, to visit the Institute or to work together on problems of mutual interest. Laurits Bjerrum was successful in implementing this policy, to which the

Institute's guest book bears witness, showing that a total of 1,974 guests had visited the Institute during Bjerrum's 22 years as director, and no less than 100 foreign specialists had been resident at the Institute for a stay of two months or more.

In the years immediately following the creation of the NGI, Laurits found at Imperial College London a valuable source of ideas and stimuli, especially in the application of effective stresses to stability

problems whose methods of solution were then the most advanced at the time. Complementing the theoretical ideas, he found well-developed triaxial testing equipment and interesting field instrumentation techniques, such as the BRS vibrating wire stress tester. In 1957 NGI founded Geonor

to manufacture and sell instruments and equipment for geotechnical laboratories. The BRS vibrating wire stress tester has since formed the basis of many of NGI Geonor's field instruments.


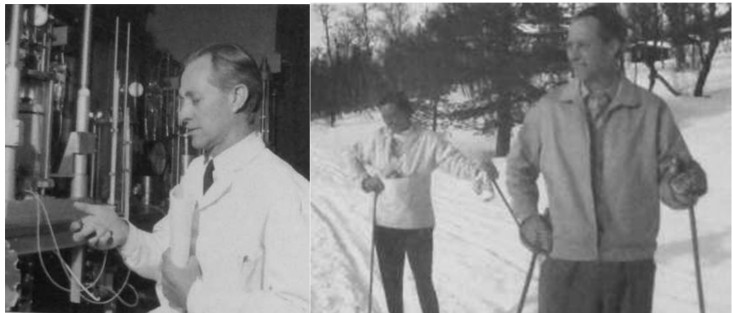

**Figure 7. (Left) Laurits Bjerrum in the NGI laboratory. (Right) Laurits Bjerrum and his wife Gudrun cross-country skiing in Rau Land, Norway. (Lersbryggen et al., 2003)**

### 5  Travel & international relations

**5.1 Alec Skempton**

Bjerrum also attended the 1950 Symposium on Shear Strength in London in 1950, where he presented an important paper on the shear strength of soils, which was subsequently published in 1951 in the journal Géotechnique, (Bjerrum, 1951). That paper attracted the interest of Alec Skempton and Alfred Bishop, who were studying the same subject at Imperial College in London.

At that congress Laurits was able to extend his contacts with leading experts in soil mechanics from all over the world, and, in many cases, to meet again the people he had met at the Rotterdam congress. From these personal relationships important friendships developed. One of them is well documented, such as that of Alec Westley Skempton.

**5.2 Karl von Terzaghi**

The relationship with Terzaghi began with a letter that Karl Terzaghi wrote to Laurits Bjerrum in 1954, after reading his article on the properties of Norwegian clays. The paper, entitled "Stability of natural



slopes in quick clays", (Bjerrum, 1954b), was presented to the European Congress on Slope Stability, held in Stockholm, Sweden, in 1954.

Shortly before, in October and December 1953, two landslides occurred, at Bekklaget and Ullendaker, both in Norway, on typical Norwegian clays, and their study was one of the first NGI papers. The Norwegians had a long history of slides in such clays, with records of some slides dating back to 1345, so it is not surprising that popular contact with the phenomenon gave it a name, kvikkleire meaning quick clays. It was probably Bjerrum who, in his study of these clays, first wrote down this Norwegian

word, the English translation of which, quick clays, became a reference. These clays had an anomaly in the form of a low salt content in the interstitial water, which was not typical of a clay of marine origin, and which gave them a peculiar behaviour.

Bjerrum agreed very much with Terzaghi's views, which helped a good collaboration and mutual affection to develop quickly. So much so that those who knew Terzaghi well considered that Bjerrum

was undoubtedly the last person to befriend Terzaghi before passing away.

Peck said: "*Terzaghi's relationship with Bjerrum began later than Skempton's, Casagrande's or mine. Even so, their relationship was remarkably close. They had an instinctive and total appreciation of each other (...) Perhaps a fundamental reason for their mutual appreciation and regard was that they were both, first and foremost, engineers, albeit with an outstanding scientific curiosity and aptitude*" (Peck,

280 1984).

### 5.3 Ralph Peck, Arthur Casagrande and their visits to the USA

In 1953 Bjerrum attended the 3rd International Congress of Soil Mechanics and Foundation Engineering in Zurich, Switzerland, where he again met Alec Skempton and, while they were talking,

Ralph Peck, an American civil engineer and another of the leading figures in soil mechanics, introduced himself to Skempton and Skempton introduced Laurits Bjerrum. After the introductions almost immediately, the three of them agreed to have dinner that evening. Laurits, who knew Zurich very well where he worked from 1947 to 1951, chose a quiet restaurant by Lake Zurich, and during this dinner, a good personal relationship with the then unknown Ralph Peck was initiated and the relationship

between Bjerrum and Skempton was strengthened. The congress started in Zurich, that included four days of technical visits, and ended in Lausanne. Among the technical visits, as mentioned above, was a



visit to the Marmorera Dam, which was well known to Bjerrum from his years of work at the Research Institute of Hydraulic and Civil Engineering at the ETH Zurich.

Later, during the same congress, Bjerrum, Peck and Skempton had the opportunity to meet for dinner, this time, with Juul Hvorslev, another international figure in Soil Mechanics, who spoke to them about his research on effective stresses. As a result of this dinner, and especially encouraged by Laurits Bjerrum, Skempton decided to apply effective stress analysis to slopes in the London clays.

This is an example of Bjerrum's lifelong influence on the people around him, which he manifested throughout his life, along with his passionate enthusiasm, great intelligence, and endless capacity for

hard work.

Skempton knew right away that Laurits was the person who would lead the way in Europe in the difficult field of clay shear strength and, among many other shared interests, they also coincided with Juul Hvosrlev's dedication to good work. Bjerrum and Skempton shared the conviction of the importance of fieldwork in Soil Mechanics, and that both geology and soil history had to be considered.

His first trip to the United States was in 1956. At the Massachusetts Institute of Technology (M.I.T.) they met, among others, Ralph Peck, Karl Terzaghi and the brothers Arthur and Leo Casagrande. M.I.T. offered Bjerrum a position as a visiting professor, which led him to spend the first six months of 1957 in Boston.

After his return to Oslo, Bjerrum was visited in August 1957 by Karl Terzaghi and Arthur Casagrande

who were interested in learning about the NGI. They also took the opportunity to make a short trip along the west coast of Norway.

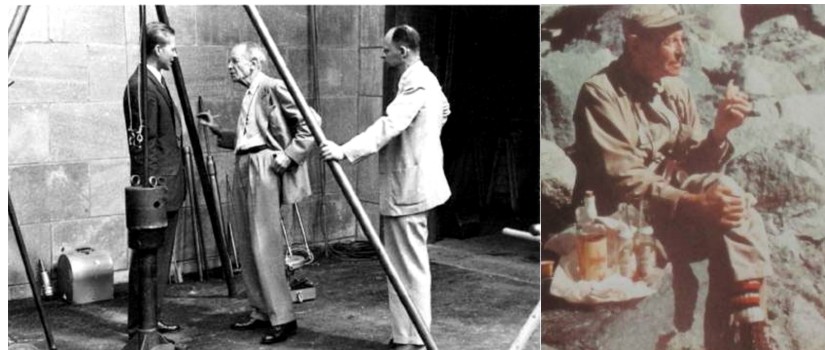

**Figure 8. (Left) Laurits Bjerrum (39), Karl Terzaghi (74) and Arthur Casagrande (55) during their visit to part of the NGI facilities. (Right) Karl Terzaghi in Loen, on the west coast of Norway. Both photos in August 1957. (Bjerrum, G., 2003)**

In the autumn of 1957, Bjerrum was invited to give several lectures at the University of Beograd in Yugoslavia. On the way he stopped in Vienna to meet with Professor Otto Fröhlich, who told Laurits that he had recently found Terzaghi's papers, which he had left behind when he left Austria in 1938 after the Nazi invasion of Austria. Among these papers, Bjerrum found technical information as well as notes and documents from the political life of that time. All these documents are now part of the Terzaghi library.

By 1960 Bjerrum was fully integrated into the NGI and was already a source of ideas and encouragement for all its members. At that time, the Institution of Civil Engineers (ICE) in London created the Rankine Lecture, to honour the memory of the civil engineer W. J. M. Rankine, who was one of the first engineers in the UK to make important contributions to soil mechanics. The first Rankine Lecture was in 1961 and the honour of giving it was given to Arthur Casagrande, (Casagrande, 1961). Bjerrum attended most Rankine Lectures in London and, taking advantage of these occasions, was usually invited to lecture at Imperial College.

In 1960, Laurits Bjerrum of NGI and Alan Bishop of Imperial College London were invited to the United States to participate in a conference. This conference, organised by the American Society of Civil Engineers (ASCE), was held in Boulder, Colorado, which became known as the Shear Strength of

Cohesive Soils and was the first of many other conferences that have been held with considerable

success over the years by the ASCE.

At that conference, Bjerrum presented a paper together with Alan Bishop (Bishop & Bjerrum, 1960). Bjerrum's participation in that conference was of great importance because it consolidated the professional links of the NGI, and of Imperial College, with the Soil Mechanics that were developing in the United States.


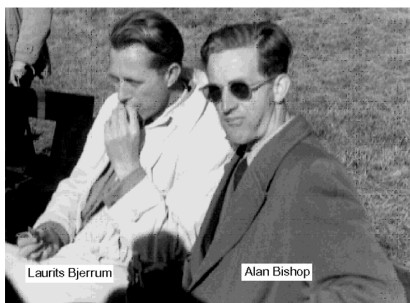

**Figure 9. Laurits Bjerrum and Alan Bishop in 1960, (Guillán Llorente, 2015)**

The usual format of these monographic congresses consisted of a series of thematic sessions with the

participation of leading figures in the field, led by a moderator. Each member of the session would read a short presentation, followed by a formal discussion among them.

"*I (Ralph Peck) was the moderator in one of the last sessions, and our group decided to encourage spontaneity. I proposed that the session discuss, informally, a specific problem: How to estimate the shear strength of sloping ground on which a dam of loose materials is built. Stanley Wilson voluntarily*

*began the discussion. He had barely said which soil tests he would carry out when Bjerrum rose from his seat, commandeered the microphone and exclaimed, "I don't agree!" He surprised not only the audience but also Stanley himself and me. A lively impromptu debate ensued that delighted the audience and detailed, better than any sober technical debate, the real points of the limitations of our knowledge. Typically, Bjerrum didn't just play the game, he made the rules and encouraged debate. The audience*

*saw what they had come to see 'big names' in action*" (Peck, 1984).

Through his teaching and lecturing, Bjerrum, who was probably more instrumental than anyone else, was bringing to the United States the soil knowledge available in Europe in the 1950s.

Bjerrum was in the United States several times, and for long periods, to visit and lecture at M.I.T. where he became first a professor and then a visiting professor in the department of Civil Engineering between

1957 and 1964.

When he visited M.I.T., most of the time he stayed at Terzaghi's house, where he soon came to be considered as one of the family, which undoubtedly strengthened their relationship, both professionally and in human terms.

Evidence of this close relationship over the years was Terzaghi donated all his papers, articles, and

records of experiments, both from his time in Europe and in the United States, to the library established in his honour by Laurits Bjerrum at the NGI in Oslo. The Terzaghi Library was inaugurated, in Oslo, in the autumn of 1957.

**5.4 Intensification of conferences**

Also of great importance was Bjerrum's attendance in 1961 at the 5th International Conference on Soil Mechanics and Foundation Engineering in Paris, where Skempton handed over the presidency to Arthur Casagrande and Bjerrum was named vice president representing Europe until 1965. Participation from the Norwegian delegation at the Paris Conference was significant, with most NGI engineers attending.

It should be noted that shortly afterwards, on 25 October 1963, Karl Terzaghi passed away. Terzaghi

was the main supporter of Bjerrum in the USA. Proof of this are the words of Nilmar Janbu "*Grosso modo and in a simplified form it can be said that geotechnics in Norway before 1950 was led by the engineer Sverre Skaven-Haug of the state railways. Before 1950, international geotechnics was mainly in the hands of Harvard University and M.I.T., with Casagrande and Terzaghi. After 1950 it became increasingly NGI and Imperial College London, with Bjerrum and Skempton as prominent individuals*",

(Janbu, 2003).

In 1964 he was elected as a Corporate Fellow of the Institution of Civil Engineers in the UK. In the same year, on Good Friday 1964, a major earthquake occurred in Alaska. The Alaska District Corps of Engineers appointed a panel of experts consisting of Thomas (Tom) M. Leps (an engineering geologist), Laurits Bjerrum and Ralph Peck (Hanson, 1984), to help coordinate and review the work being done on

the landslides associated with the earthquake. This was the first time that Bjerrum and Peck had worked

together (remember that he had met him in 1953 at the Zurich congress), and the relationship was so satisfactory to both that they agreed, if possible, to find a project in the future where they could be associated again.

Bjerrus and Peck found several projects to work on together. They worked as consultants on the breaching of the Cannelton and Uniontown dam cofferdams on the Ohio River, also on the Oslo underground, on the James Bay hydroelectric development in Canada and, for almost 6 years, on a dam project in the Dead Sea. On that project, the contractor was sued by the dam owner, who argued that the completed dam did not meet design specifications. The contractor hired a consultant committee consisting of Ralph Peck and Laurits Bjerrum, with two collaborating engineers, Thomas (Tom) M. Leps for Peck and Kaare Høeg for Bjerrum. The committee sent Høeg to the Dead Sea for a month to conduct in situ permeability tests, using a modern piezometer designed by Geonor, NGI's instrumental development company. It was found that the hundreds of tests had most likely caused a systematic hydraulic fracturing of the core and thus overestimated the permeability of the dike core. This hypothesis was subsequently verified by in situ testing in the clays of Fodrnebu, Oslo, and led to the development of a technique to measure the lateral stress of hydraulic fracturing in soft clays. The contractor was very pleased with the findings because it significantly strengthened its position in the claim.

During the afternoons after work, Bjerrum and Peck liked to debate the frontiers of knowledge: "*secondary consolidation, deformation energy, diagenetic linkages, liquefaction. He was the innovator, the proponent, and I (Peck) was the objector, the conservative. Each of us expressed his point of view and by defending them we brought the discussion as close to reality as possible*" (Peck, 1984).

From 1965 to 1969, Bjerrum was the president of the International Society of Soil Mechanics and Foundation Engineering, taking over the baton from Arthur Casagrande at the 6th International Conference on Soil Mechanics and Foundation Engineering, held in Montreal, and handing it over to Ralph Peck at the 7th Conference held in Mexico City. At the same time, he was a member of the steering committee of the International Society for Rock Mechanics during the years 1962 to 1966, when the Austrian Leopold Müller held the presidency.

By the time of the Montreal Conference in 1965, Laurits had become a regular visitor to the United States, and had several classes scheduled by William (Bill) Lambe at M.I.T. Usually, when he visited M.I.T. he also visited Peck at the University of Illinois, which made both families more than just



acquaintances. An example of this was the trip the two couples took together on a weekend trip during the Montreal Conference. At that time of the year, the leaves of the maple trees were at their brightest red. Bjerrum was fascinated by the landscape, so much so that he said that anyone who has not seen Canada when the leaves change colour cannot understand why the maple leaf is Canada's national

symbol.

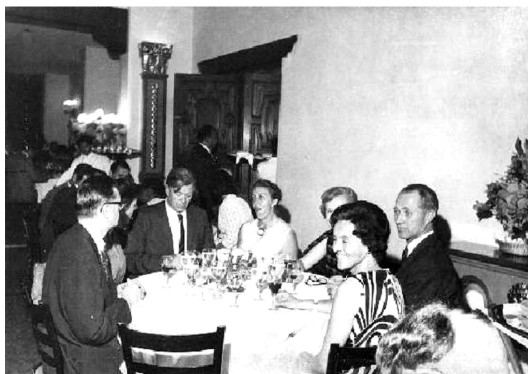

**Figure 10. (Clockwise from left) Ralph Peck (57), Alec Skempton (55), Nancy Skempton, Marjorie Peck, Laurits Bjerrum (51) and Gudrun Bjerrum at the 7th International Conference on Soil Mechanics and**

**Foundation Engineering in Mexico City 1969. At this Conference, Bjerrum handed over the chairmanship to Peck, (Guillán Llorente, 2015)**

At the same Conference in Montreal, as it was the first conference held after the death of Karl Terzaghi, a special session was held in his honour. In that session, Laurits Bjerrum explained the reason for the

Terzaghi Library in Oslo. Years later, in 1966, Bjerrum was awarded by the ASCE the honour of delivering the 3rd Karl Terzaghi Lecture in memory of his friend and founder of Soil Mechanics. Bjerrum began his lecture by saying: "*I would be dishonest if I did not admit that I am extremely happy about this unexpected recognition for my work. But I must also admit that, as the recipient of this award, I consider myself only as the official representative of a group of colleagues and good friends in*

*Oslo, with whom I have cooperated so intimately that it is impossible to differentiate their contribution*

*from mine*". In this paper Bjerrum laid the foundation for understanding the shear strength of overconsolidated clays and overconsolidated clay shales (Bjerrum, 1967).

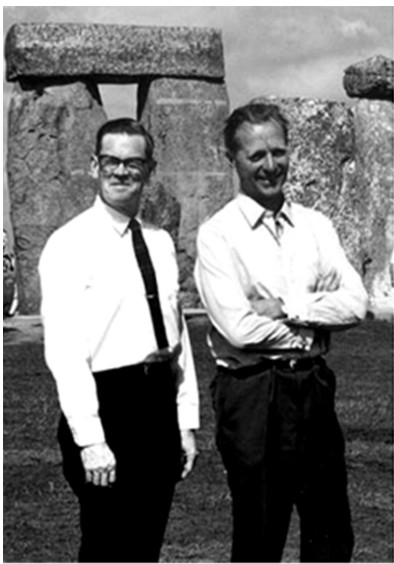

**Figure 11. Ralph Peck (56) and Laurits Bjerrum (50) during a visit to Stonehenge, England, in 1968,**

**coinciding with a meeting on the Dead Sea Dyke, (Guillán Llorente, 2015)**

As far as Bjerrum and the UK were concerned, this period of expansion and leadership culminated in the 7th Rankine Lecture in 1967, which Bjerrum had the honour of giving, where an original idea, "Engineering geology of normally consolidated Norwegian marine clays and its relation to the seating

of buildings", was masterfully presented and supported by consistent field and laboratory work. A curiosity about this Rankine Lecture is that the presentation was given by Alec Skempton, who years earlier, in 1964, at the 4th Rankine Lecture had the honour of giving, with the gratitude speech delivered by Bjerrum. Indeed, the relationship between Bjerrum and Skempton went beyond the professional to a familial relationship. In 1972 Bjerrum attended the 5th European Congress on Soil Mechanics and

Foundation Engineering in Madrid. Bjerrum's paper entitled "Earth pressures in flexible structures" was

a state-of-the-art presentation focusing on a better understanding of the basic phenomena governing the behaviour of the soil surrounding a shored excavation (Bjerrum et al., 1972.

He also presented another paper, at the same congress, describing a new method for determining the lateral pressure in normally consolidated clays, (Bjerrum & Andersen, 1972).

**6  Passing away**

On the day of his death, 27 February 1973, Laurits was in London on the eve of the Rankine Lecture, where he had been invited to introduce the laureate William (Bill) Lambe and to give lectures at Imperial College on subsurface research in the North Sea and state of marine geotechnical engineering.

At the time of his death, Bjerrum was on the advisory board of the James Bay Hydroelectric Project, the

largest project in the world, along with Ralph Peck. The project, which consisted of more than 220 dams and dikes, with a total length of more than 140 km. From the beginning, it was subjected to strong economic pressures. But the treatment of the foundation, insistently advised by Bjerrum, was freed from most of the economic cuts, (Peck, 1980).

In 1973, the year of Bjerrum's death, the NGI had a permanent staff of 95 people. His replacement as

head of NGI was the Norwegian civil engineer Kaare Høeg, (Lacasse, 2000). One of the last things Bjerrum did in the NGI was to suggest the symbol that was adopted as the official emblem of the Institute. The emblem was originally used at the geotechnical congress held in Oslo in 1967.

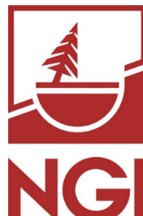

**Figure 12. Official Emblem adopted by the NGI in 1973, (Guillán Llorente, 2015)**


He was survived by his wife Gudrun and his three children Chresten Bjerrum, Annette (Bjerrum) Lerbryggen, and Susanne (Sanne) (Bjerrum) Møller.

Following Bjerrum's death, Skempton wrote to Gudrun Bjerrum: "I will always remember Laurits as one of my dearest friends and the most important person in Soil Mechanics after Terzaghi. My life was

happier and more interesting thanks to having met him".

## 7  Awards and acknowledgements received

During his lifetime, Bjerrum received many honours and awards, such as the aforementioned Karl Terzaghi Lecture of the ASCE in 1966 and the Rankine Lecture of the ICE in 1967. He was a member of the Danish and Norwegian Academies of Technical Sciences; Fellow of the Royal Norwegian

Society of Science and Letters; Honorary Fellow of the Norwegian Geotechnical Society; Corresponding Fellow of the British Institution of Civil Engineers; Honorary Doctor of Science from Loyola College, Baltimore in 1965; Sam Eyde Award for outstanding work in Civil Engineering in Norway in 1966; Corresponding Member of the Institute of Civil Engineers of Venezuela. In 1971 Laurits received the Terzaghi Award, presented to him by Karl Terzaghi's widow, Ruth, at the Purdue

conference in Lafayette, Indiana, for his and NGI's work on embankments in soft clays.

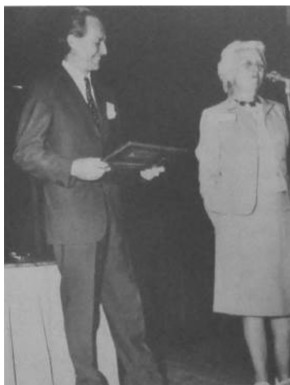

**Figure 13. Ruth Terzaghi at the presentation of the Terzaghi Prize to Laurits Bjerrum, 1971, (Bjerrum, G, 2003)**

## 8 Conclusions

Bjerrum, like Terzaghi, Casagrande and Skempton, was orphaned at a very young age. Despite the absence of his father, the family had a comfortable financial position, which enabled him to approach life with a certain carefree attitude. He had a restless nature and a love of the outdoors, so it came as no surprise that he enjoyed field research, or, as he used to say, "drilling the province", referring to the province of Jutland, a large region of Denmark.

He combined his university studies with work at the Harbour Works and Foundations Laboratory, which enabled him to learn to work methodically, a fact he later applied assiduously at the NGI.

One fact to take into account is that Bjerrum lived through the Nazi occupation of Denmark, which conditioned him against the misuse of authority. This factor was reflected in the open structure he established at the NGI from its inception.

His research interests led him to review all available publications on slope slides, where he discovered that Robert Haefeli was working on similar topics, including snow mass movements, at the ETH in Zurich. The subject of reptation bore some resemblance to snow mass movements, which was part of the "snow mechanics" Haefeli was investigating. Immediately after the end of the Second World War, Bjerrum left Denmark and settled in prosperous Switzerland. The circumstances and Haefeli's health

problems meant that Bjerrum took over some of his duties, including the writing of expert opinions and various papers at the laboratory in Zurich, which was a formative addition as a researcher and populariser. At that time, the loose material dams, in which Bjerrum was extensively involved in various parts of Switzerland, were of great interest. This enabled Bjerrum to become familiar with soils that were very different from those of his native Denmark, and at the same time to broaden his

knowledge of glacial and post-glacial soils.

In 1947, the ETH Zurich was a leading centre for Soil Mechanics in Europe. Its director Eugene Meyer-Peter is listed as a registered member in the proceedings of the 1st International Conference on Soil Mechanics and Foundation Engineering at Harvard, although he is listed as an absent member. As a registered member, he had access to all the documentation of the congress, which allowed him to align

ETH with the principles of Soil Mechanics led by Terzaghi. After the Rotterdam Conference in 1948, Zurich was the host city for the third International Conference in 1953, hosted by the ETH.

This period (from 1948 to 1953) was a great technical and organisational effort for the ETH and for Bjerrum with the consolidation of the principles on which the NGI would later be based.

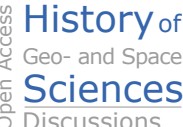

Bjerrum's attendance at the Rotterdam Conference is in line with this approach adopted by the ETH. At
the Rotterdam Conference, Bjerrum met Alec Skempton. Subsequently, Bjerrum attended the 1950
Symposium on Shear Strength in London as a member of the ETH, which allowed him to extend his
international contacts and to see people he had met at the Rotterdam Conference, in particular,
Skempton.

By the time of the Zurich Conference in 1953, Bjerrum was already settled in the NGI. It was at that
conference that Bjerrum had the opportunity to meet Peck, thanks to Skempton.

The following year, in 1954, Terzaghi wanted to meet him. No information has been found to indicate
when Bjerrum met Casagrande personally. The earliest available reference is to Bjerrum's trip to the
United States in 1956.

Regarding the understanding of the basic principles of clays in 1953, Laurits Bjerrum, was involved
with an investigation into the peculiarities of some post-glacial clays that were very common in
Norway. These clays presented the peculiarity that had a certain resistance when pressed with the
finger, and yet, a small disturbance produced a radical change in their consistency. Moreover, if the clay
was remoulded, the change in its resistance was even greater, partially transforming it into a liquid
similar in appearance to a heavy oil. This property is the origin of its name: Quick Clay. These clays
were popularly known in Norway as Kvikkleire, the closest English translation of which is quick clay.
Under the microscope, Bjerrum observed that the clay was composed of flake-like mineral particles
with water-filled pores between the flakes. In his detailed analysis, he found fragments of shells and
microfossils of species still found today in the great Arctic oceans, which attributed a marine origin.

Subsequent epirogenic movements caused the former glacial clay deposits to rise above seawater level,
forming large areas in Scandinavia and in the north of the American and Asian continents. The rise
above sea level led to a change in environmental circumstances, e.g., exposure to freshwater flows that
resulted in a gradual shift from salt water, initially confined in the clay pores, to fresh water, resulting in
a low salt concentration in the interstitial water. This leaching was responsible for the peculiar
properties of the quick clays, mainly due to the low activity of the clay minerals that condition the liquid
and plastic limits to be abnormally low. Normally consolidated Norwegian clays showed a linear
increase of undrained shear strength with depth, which could be expressed by the ratio of shear strength
to the effective overburden pressure w/w. The salt leaching was accompanied by a reduction of the



plasticity index. This led to the conclusion that the shear strength decreased with decreasing salt concentration.


**Data availability.** The research data of this paper can be found in the references. The doctoral thesis of Gonzalo Guillán (2015) is also useful.

**Author contributions.** All the authors contributed to the work. GGL was the author of the original
doctoral thesis; BMM, ALG and RGA wrote and reviewed the manuscript and translated it into English.

**Competing interests.** The authors declare that they have no conflict of interest.

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
