# Peer review of "Contribution to the knowledge of early geotechnics during the 20th century: Laurits Bjerrum"

_History of Geo- and Space Sciences, 2024_

## Author Response (AR1)

**Author's response**

Anonymous Referee #1

*I think it is very appropriate to make an investigation regarding the life of one of the most relevant researchers in geotechnics such as Bjerrum.*

*In my opinion, it describes properly his life, emphasizing most important aspects like his childhood but, also key contacts and main innovations in its area.*

*The manuscript fits the content of the journal, is well written, properly organized and fluent language.*

*Therefore, I think it is ready to be published.*

Author's response to Anonymous Referee #1,

We would like to thank the reviewer for their valuable comments on the revised manuscript.
We are pleased that the reviewer believes that there is value in our work.

Anonymous Referee #2

*The article provides an excellent overview of the life and activity of one of the founding fathers of soil mechanics and geotechnical engineering as we know it today. The clear and concise style and the technical elements presented to the required level of detail combine in a very useful and informative contribution.*

*Specific comments: The authors should introduce a short section presenting details related to the historical methods used in compiling the article and highlighting the original contributions of the authors.*

*Technical corrections:*

*In general, avoid adding an article to ETH or NGI (e.g. "at the ETH" or "of the NGI"). This correction is relevant for Lines 99, 112, 229, 293, 310, 324, 338, 501, 511, 516, 517, 519, 521, 524.*

*Line 19: Replace "was" with "were" at the end of the line*

*Line 40: Replace "prankswith" with "pranks with" by introduciung the required blank space between the words*

*Line 64: Replace "department although initially, it consisted of him alone." with "department, which initially consisted of him alone."*

*Line 91: Replace "slope slides" with "landslides"*

*Line 92: Replace "Research Institute for Hydraulic Engineering and Civil Engineering" with "Research Institute for Hydraulic and Geotechnical Engineering (VAWE, its German acronym)" for a more correct translation of the original german name Versuchsanstalt für Wasser- und Erdbau.*

*Line 93: Replace "(ETH its German acronym)" with "(ETH, its German acronym)", by adding the required comma after ETH. Add full stop after the braket and start a new sentence: "He was particularly interested..."*

*Line 101: Replace "joined" with "came to", to avoid repetition of verb "join" in the same sentence.*

*Line 102: Replace "Research Institute for Hydraulic Engineering and Civil Engineering" with "VAWE", which was already introduced in line 92.*

*Line 103: Replace "soil mechanics and hydraulics laboratory." with "geotechnical department" for a more correct translation of the original german name - "Erdbauabteilung."*

*Line 123: The statement "with whom he became a professor at the ETH" is confusing. Please clarify.*

*Line 124: Replace "the Zurich Research Institute" with "VAWE in Zurich". Replace "dams of loose materials" with "earthfill dams"*

*Line 125: Replace "loose material dam" with "earthfill dam"*

*Line 126: Replace "loose material dam" with "earthfill dam"*

*Line 130: Replace "constrution laboratory" with "construction site laboratory"*

*Line 135: Remove full stop before citation.*

*Line 140: Replace "placed on site" with "used on site for the Marmorera dam"*

*Line 141: Replace "field laboratory" with "construction site laboratory"*

*Line 142-143: Replace "an almost elastic behaviour" with "the excellent behaviour". More context would be required to state that the response is elastic.*

*Line 143: Replace "loose material dam" with "earthfill dam"*

*Line 156: Did you mean "core of NGI" when using "germ of NGI"?*

*Line 176: Replace "EHT" with "ETH"*

*Line 177: How do you define the "geotechnical role" of Norway? Adjust wording accordingly.*

*Line 178: Replace "was" with "were"*

*Lines 220-224: The first two sentences are intricate and repetitive. Suggestion for reformulation: "During this time Bjerrum completed his research for his doctoral thesis, "Theoretical and experimental investigations of shear stresses in soils", (Bjerrum, 1954a), with which he obtained his doctorate in technological sciences from ETH Zurich in 1954, being supervised by Prof. Meyer-Peter, E. and Prof. Haefeli, R. This confirmed the beginnning...."*

*Lines 239-240: Replace "the application of effective stresses to stability problems whose methods of solution were then the most advanced at the time" with "the application of the*

*effective stress method to slope stability problems, which was the most advanced at the time."*

*Line 242: The terminology vibrating wire stress tester is confusing. Is this a strain gauge?*

*Line 271: Replace "interstitial water" with "pore water"*

*Line 286: Replace "and" with "while" for more clarity and less repetition.*

*Line 290: Remove "that"*

*Line 292-293: Replace "Research Institute of Hydraulic and Civil Engineering" with "VAWE" or alternatively with the correct translation "Research Institute for Hydraulic and Geotechnical Engineering"*

*Line 396: Replace "instrumental development company" with "company for equipment development"*

*Line 396-397: The statement is confusing: "It was found that the hundreds of tests had most likely caused a systematic hydraulic fracturing of the core and thus overestimated the permeability of the dike core". Please reformulate.*

*Line 446-448: The statement is confusing: "the presentation was given by Alec Skempton, who years earlier, in 1964, at the 4th Rankine Lecture had the honour of giving, with the gratitude speech delivered by Bjerrum". It is not clear what Skempton's role was in the 7th rankine lecture. For sure, Bjerum proposed the vote of thanks for Skempton's Rankine Lecture in '64. Please reformulate, clarify and use correct terminology (i.e. "vote of thanks" rather than "gratitude speech")*

*Line 452: Replace "shored" with "supported"*

*Line 493: Replace "drilling the province" with "drilling in the province"*

*Line 500: Replace "slope slides" with "landslides"*

*Line 502: The formulation is unclear "subject of reptation". Please reformulate.*

*Line 507: Replace "loose material dams" with "earthfill dams"*

*Line 531: Replace "that had" with "of having"*

*Line 537: Replace "found" with "observed" to avoid repetition.*

*Line 538: Replace "marine origin" with "marine origine to the quick clay".*

*Line 543: Replace "interstitial water" with "pore water".*

Author's response to Anonymous Referee #2,

We are pleased that the reviewer believes that there is value in our work. Thank you for your comments on the manuscript and we appreciate the effort you made in carrying out such a thorough review. We believe that the comments are very valuable and have helped us a lot to improve the quality of the manuscript.

Following your comments, we have included a new section, Methodology, indicating what has been the process to establish the chronological order of the different events and data, and thus determine the complete biography in an orderly manner.

Similarly, we have made all the changes indicated.

Editor

*As Topical Editor I have only two minor formal items to note:*

*1. The left-hand photo of Fig. 5 is of low quality, the faces are hardly discernible. Is this photo really necessary?*

*2. in Line 389 it should probably read Bjerrum not Bjerrus.*

Author's response to Editor

Dear Editor,

1. There is no need for this photo.

2. This is a spelling mistake. The correct term is Bjerrum.